# Algorithm For Concept Extrapolation: Diverse Generalization via Selective Disagreement

## Abstract

Standard deep learning approaches often struggle to handle out-of-distribution data, especially when the distributional shift breaks spurious correlations present in the training data. While some approaches to handling spurious correlations under distributional shift aim to separate causal and spurious features without access to target distribution data, such approaches typically rely on labeled data from different domains or contingent assumptions about the nature of neural representations. Existing methods that do use unlabeled target data make strict assumptions about the target data distribution. To overcome these limitations, we present the Algorithm for Concept Extrapolation (ACE). Formulating diverse generalization as a *semi-supervised* learning problem, ACE applies a psuedo-labeling approach, learning from pseudo-labels that flexibility incorporates assumptions about the target distribution. As a result, we find that ACE is more robust to variations in the target distribution than prior methods. We also demonstrate an initial application of ACE to alignment, finding that ACE can be made to improve goal misgeneralization. Overall, we are exciting about the opportunities ACE opens for learning diverse generalizations of under-specified labels on new domains.

## 1 Introduction

Standard deep learning approaches often struggle to maintain high performance under distributional shifts (Hendrycks & Dietterich, 2019). Models are particularly sensitive to distributional shifts that "break" the correlation (present on the training distribution) between "spurious" and "ground truth" features (e.g. classifying huskies based on the presence of snow (Ribeiro et al., 2016), or images based on texture rather than shape (Geirhos et al., 2022). Spurious correlations are problematic when the spurious feature is simpler than the ground truth feature - due to the simplicity bias of neural networks trained with stochastic gradient descent (Shah et al., 2020; Mingard et al., 2020), neural networks tend to to learn the simpler spurious feature rather then the ground truth feature, *even if* the spurious feature is less predictive (Hermann et al., 2024).

Spurious correlations breaking under distributional shift can cause degradation in performance in safety critical domains (e.g. medical imaging (Zech et al., 2018)) and create algorithmic bias if a protected attribute is correlated with particular features or outcomes ((Dressel & Farid, 2018; Obermeyer et al., 2019)). Current issues with aligning foundational models to human intent (Ouyang et al., 2022; Askell et al., 2021), such as reward model over-optimization (Gao et al., 2022), sycophancy (Sharma et al., 2023; Denison et al., 2024), weak to strong generalization (Burns et al., 2023), and measurement tampering (Roger et al., 2023) can be cast as spurious correlations (typically between the reward process and the object of reward) breaking under distributional shift (see Appendix D for an extended analysis of relevance to alignment).

To address spurious correlations breaking under distribution shift, some work aims to learn classifiers that track ground truth features using only labeled training data. In domain generalization, methods rely on labeled data sampled from different domains, attempting to learn features that are robust across the training domains such that they generalize to the test domain ((Arjovsky et al., 2020; Sagawa et al., 2020; Krueger et al., 2021)). However, data sampled from multiple domains is not always available, and when it is, standard empirical risk minimization is often competitive

(Gulrajani & Lopez-Paz, 2020). More recent work attempts to learn an ensemble of diverse classifiers with *representational* diversity, with metrics for representational diversity including direction of classifier gradients (Teney et al., 2022) and layer at which features can be learned (Tiwari & Shenoy, 2023). While these methods do not require labeled data from different domains, they rely on particular (and possibly contingent) assumptions about neural network representations, and must anticipate the target distribution by learning all plausible generalizations.

Access to unlabeled target data makes the problem of learning diverse generalizations significantly more tractable, as the space of classifiers that behave differently on the target distribution is much smaller than the space of classifiers that disagree on arbitrary out-of-distribution data (see Sec **??** for more a formal argument). Domain adaptation methods leverage unlabeled target data, but most algorithms focus on more benign distributional shifts where e.g. the background of all image changes change from one setting to another, rather than distributional shifts that break spurious correlations (Saenko et al., 2010; Wilson & Cook, 2020).

To leverage unlabeled target to overcome spurious correlations, we propose learning an ensemble of classifiers with *predictive* diversity, finding classifiers that achieve low risk on the source distribution while selectively disagreeing on target distribution instances that break spurious correlations. Two concurrently developed methods, DivDis(Lee et al., 2023) and D-BAT (Pagliardini et al., 2022) propose similar methods, regularizing classifiers towards statistical independence (DivDis), or maximal disagreement (D-BAT) on the target distribution. However, both methods make assumptions about the proportion of correlation-breaking instances on the target distribution. In our experiments, we show that the performance of DivDis significantly degrades when these assumptions are violated.

To overcome these limitations, we introduce Algorithm for Concept Extrapolation (ACE) [1]. Motivated by the framing of learning diverse classifiers as a semi-supervised learning problem, ACE uses a *pseudo-labeling* (Lee, 2013) approach, generating pseudo-labels on the target distribution for each classifier in proportion to the expected joint feature distribution. To account for noise and uncertainty in the pseudo-labeling process, ACE applies exponential smoothing, placing higher weight on more confident predictions.

On a broad set of image spurious correlation benchmarks, we find ACE achieves comparable performance to DivDis on target distributions with statistically independent features, and substantially *superior* performance on distributions with small or large proportions of instances that break spurious correlations. We also find ACE can improve the generalization of deep reinforcement learning agents, with our ACE-Agent successfully completing 16% more levels in the CoinRun goal misgeneralization problem (Langosco et al., 2023; Shah et al., 2022) than baseline approaches.

In section 2, we formulate learning diverse generalization as a semi-supervised learning problem, and explain how optimal classifiers DivDis and D-BAT diverge from optimal classifiers under our formulation. In section 3, we introduce ACE as a pseudo-labeling approach, present an alternative "Top-K" formulation of ACE, and detail ACE's exponential smoothing. In section 4 we present experiments and results.

## 2 FORMULATION AND MOTIVATION

### 2.1 FORMULATION

Suppose we have a source distribution $p_S(x, y_1, ..., y_H)$ over inputs $x$ and multiple labels $y_1, ..., y_H$, with each label corresponding to a different *feature* of the input (e.g. animal, background). In the case of *complete* (positive) spurious correlation, the class labels are always equal. Now suppose we have a target distribution $p_T(x, y, .., y_H)$, where the joint class distribution shifts, such that there are some instances on which the class labels disagree. We can the proportion of instances with disagreement labels the *mix rate* [2]:

$$r = 1 - p(y_1 = y_2 = \cdots = y_H) \tag{1}$$

Our objective is to jointly learn a set of hypotheses $h_1, ..., h_N$ that achieve low risk across the source *and* target distribution, *without labeled access to the target distribution*.

---

[1] patent pending

[2] The term mix rate is somewhat of a historical accident, inspired by the term "mix ratio" in (Lee et al., 2023), but in our case referring to the target rather than source distribution

Formally, letting $\mathcal{R}_p$ denote the empirical risk under the data distribution $p(x, y_1, ..., y_H)$, $L : X \times Y \to \mathbb{R}$ denote a loss function, we have the following risk minimization problem:

$$\mathcal{R}_p(h_1, ..., h_H) = \mathbb{E}_{(x,y_1,...,y_H) \sim p(x,y_1,...,y_H)} \left[ \sum_i L(h_i(x), y_i) \right]$$

$$\min \mathcal{R}(h_1, ..., h_H) = R_{p_S}(h_1, ..., h_H) + R_{p_T}(h_1, ..., h_H) \tag{2}$$

On this view, handling spurious correlations under distribution shift becomes a (very particular) *semi-supervised learning problem*, with class labels corresponding to ground-truth and spurious features. And, as is generally true in semi-supervised learning, the success of methods will depend on whether and to what degree the assumptions made by the method hold true on the actual data distribution.

## 2.2 Implicit Mix Rate Assumptions

The two existing methods most applicable to our problem formulation are DivDis (Lee et al., 2023) and D-BAT (Pagliardini et al., 2022). Both train an ensemble of classifiers to achieve low risk on a source distribution, while maximizing some diversity loss on an unlabeled target distribution. However, methods make implicit assumptions about the joint class distribution on the target data which do not always hold. We summarize each method in turn, and explain these implicit assumptions.

**DivDis** (Lee et al., 2023) incentives diverse generalizations by minimizing the *mutual information* of the the class predictions of the hypotheses in the ensemble. Formally, for two hypotheses $h_1, h_2$, the mutual information loss is given by

$$\mathcal{L}_{\text{MI}}(h_1, h_2) = D_{\text{KL}}(p_{h_1,h_2} || p_{h_1} \otimes p_{h_2}) \tag{3}$$

with $p_{h_1,h_2}$ the joint distribution over hypothesis class predictions and $p_h$ the marginal distribution for a single hypothesis. Note that mutual information corresponds to statistical independence: variables are statistically independent if and only if they have no mutual information. In the case of feature labels, statistical independence corresponds to a mix rate of *mix rate $r = 0.5$* (exactly half of the instances have disagreeing labels). But in general we should not expect each class to be statistically independent on the target set. For example, we could have distribution shift where the features are completely *negatively* correlated (i.e. mix rate of 1.0), in which case minimizing mutual information would not correspond to achieving low risk.

**D-BAT** (Pagliardini et al., 2022) incentivised diverse generalization by rewarding *disagreement* among the set of hypotheses. Formally, assuming binary feature labels and two hypotheses $h_1, h_2$, the agreement loss is given by the negative log likelihood of disagreement:

$$\mathcal{L}_{\text{A}}(h_i, h_j) = -\log\left(p_{h_1}^{(0)} \cdot p_{h_2}^{(1)} + p_{h_1}^{(1)} \cdot p_{h_2}^{(0)}\right) \tag{4}$$

This disagreement loss (by construction) is minimized by maximally disagreeing hypotheses, but we should only expect the optimal hypotheses to maximally disagree if the feature labels are exactly negative correlated ($r = 1$) which again, we should not expect to hold in general.

The implicit assumptions of methods need not *exactly* hold in order for them to be effective. Both DivDis and D-BAT might serve as effective *regularizers* against the simplicity bias, even if minimizing the regularization loss would produce pathological hypotheses. Ideally though our a diversity loss should be minimized by hypotheses that minimize empirical risk to avoid pathological cases where the assumptions of our method strongly diverge from the actual data distribution.

## 3 ACE: Algorithm For Concept Extrapolation

We now describe our Algorithm for Concept Extrapolation (ACE). Our goal is to design an algorithm that flexibly incorporates assumptions about the target distribution.

With the diverse generalization structured as a semi-supervised learning problem, we can take inspiration from a tried-and-true approach to semi-supervised learning: pseudo-labeling (Lee, 2013) . Pseudo-labeling approaches "harden" model outputs on unlabeled data, and train the model against

those hardened outputs, effectively penalizing high entropy predictions on the assumption that class condition distributions are separated by low density regions (Chapelle & Zien, 2005; Grandvalet & Bengio, 2004). Naive approaches to pseudo-labeling would likely be counterproductive in mitigating simplicity bias, as pseudo-labels would enforce the model's existing predictions. But given access to the joint distribution of feature labels (and in particular, the rate at which feature labels disagree), we can produce pseudo-labels based on thresholds set on the classifiers to align with the expected target mix rate.

We first consider the case of two hypotheses $h_1, h_2$, and binary class labels $Y_1 = Y_2 = \{0, 1\}$. For each instance $x$, we assign pseudo labels $y'_1, y'_2$ based on the joint probability of disagreement in either direction, and ignore instances will no strong disagreement in either direction

$$(y'_1, y'_2) = \begin{cases} (0,1) & \text{if } p^{(0,1)}_{(h_1,h_2)} \geq t^{(0,1)} \\ (1,0) & \text{if } p^{(1,0)}_{(h_1,h_2)} \geq t^{(1,0)} \\ \emptyset & \text{else} \end{cases} \tag{5}$$

where $t^{(0,1)}, t^{(1,0)}$ are set per mini batch $\mathbf{x}$ to align with the assumed feature probabilities $p^{(0,1)}, p^{(1,0)}$, and $\emptyset$ denotes no label (such that $\forall x, L(h(x), \emptyset) = 0$) [3]

## 4 EXPERIMENTS

We aim to answer 6 research questions: 1) How does ACE perform on target distributions with statistically independent feature labels, 2) How does ACE perform relative to DivDis on target distribution with positively or negative correlated feature labels (low and and high mix rates), 3) How does ACE perform when the target mix rates is *higher* than the assumed mix rate, 4) How does ACE perform when the target mix rate is *lower* than the assumed mix rate, 5) How does ACE alter the the learned representations of (pre-trained) models, and 6) Can ACE mitigate goal misgeneralization in deep reinforcement learning. To answer these questions, we train various configurations of ACE and DivDis on synthetic and real world image datasets with artificially varied mix rates. We also construct a reward dataset from a deep reinforcement learning (RL) environment, using ACE to learn a reward function and subsequent training a RL agent with the learned reward.

### 4.1 EXPERIMENTAL SETUP

Unless otherwise stated, we train 2 heads on a shared Resnet50 (He et al., 2015) backbone, optimizing weights with Adam (Kingma & Ba, 2017) with a learning rate of 0.0001, weight decay of 0.0001, and source and target batch sizes of 32, over 10 epochs. As a model selection criteria (Gulrajani & Lopez-Paz, 2020), we pick models with the lowest total validation loss (source cross entropy and target diversity) (Albuquerque et al., 2019).

### 4.2 RESOLVING COMPLETE SPURIOUS CORRELATION ACROSS MIX RATES

To evaluate questions 1-4, we train various configurations of ACE and DivDis on synthetic and real world image datasets ranging in complexity and degree of simplicity bias, across artificially varied mix rates $[0.1, 0.25, 0.5, 0.75, 1.0]$. In particular, we train DivDis with a mutual information loss weight of 1 and 10, and ACE with the known global mix rate, and assumed lower bounds $0.1, 0.5$. We describe the datasets below:

**Gaussian 2D Grid**: A modified version of the toy 2D grid dataset introduced by (Lee et al., 2023), where instead of uniformly distributing data across quadrants, we sample data points from 2D gaussian distributions centered in the middle of each quadrant, with a standard deviation of 0.1, reflecting the more realistic assumption features separated by low density regions.

**FashionMNIST-MNIST**: A dataset of concatenated FashionMNIST (Xiao et al., 2017) and MNIST (Deng, 2012) images introduced by (Pagliardini et al., 2022), taking inspiration from the "dominos" format introduced by (Shah et al., 2020) (see CIFAR-MNIST below). The source distribution contains MNIST 0's concatenated with FashionMNIST tops, and MNIST 1's concatenated with FashionMNIST bottoms. The target dataset contains some instances of MNIST 0's concatenated to FashionMNIST bottoms, and MNIST 1's concatenated to FashionMNIST tops.

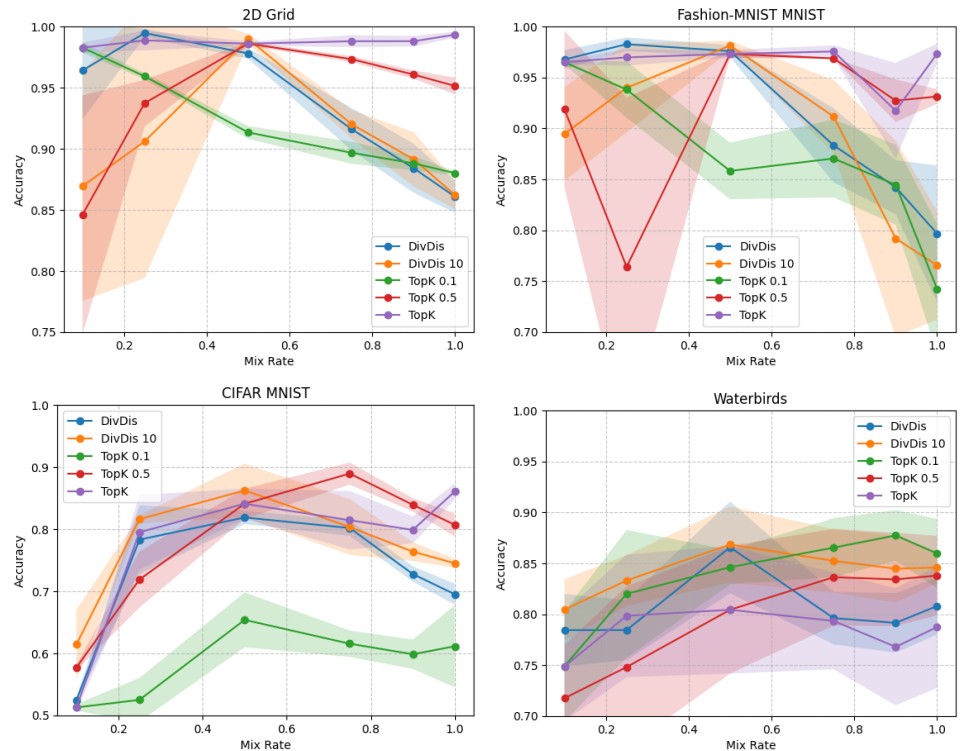

Figure 1: **Accuracy across Mix Rates** Mean and standard deviation of accuracy over 3 seeds across target mix rates for configurations of ACE and DivDis. In general, ACE outperforms DivDis on mix rates higher than 0.5, while both are fairly robust to low mix rates.

**CIFAR-MNIST**: A dataset of concatenated CIFAR-10 and MNIST images introduced by (Shah et al., 2020). The source distribution contains MNIST 0's concatentated with CIFAR cars, and MNIST 1's concatenated with CIFAR trucks. The target dataset contains some instances of MNIST 0's concatenated to CIFAR trucks, and MNIST 1's concatenated to CIFAR cars.

**WaterbirdsCC**: A modified version of Waterbirds (Sagawa et al., 2020) introduced by (Lee et al., 2023) to evaluate complete spurious correlation, WaterbirdsCC contains only images of land birds on land and waterbirds on water in the source distribution, and some instances of land birds on water and waterbirds on land in the target distribution.

Results for all the methods across all datasets and mix rates across 3 random are shown in Figure 1.

In general, we find that ACE is competitive with DivDis on mix rates of 0.5 - achieving approximately the same (albeit slightly lower) accuracy on all datasets except Waterbirds, where ACE with an assumed lower bound of 0.5 is substantially worse than DivDis, but ACE with an assumed lower bound mix rate of 0.1 only marginally so. We suspect this is due to the uneven group distribution of waterbirds - approximately three quarters of the disagreeing instances are land birds on water, meaning the assumed mix rate for waterbirds on land (0.25) overestimated the actual value (approximately 0.125).

On low mix rates, we find that ACE and DivDis are both robust on the simpler datasets (2D Grid and FashionMNIST-MNIST), while the performance of both degrade on the more complex datasets (CIFAR-MNIST and Waterbirds). Notably, DivDis with auxiliary weight of 10 suffers losses even on the simple datasets, suggesting that DivDis does not approximate optimal solutions in low mix rate regimes in general. On higher mix rates, we see a clearer divergence between ACE and DivDis, with ACE robust on all datasets while both DivDis configurations degrade substantially. This again validates our hypothesis that DivDis is ill-suited to target distributions where features are not statistically independent.

Evaluating the impact of mix rate assumptions that are higher and lower than the ground truth mix rate, we find that across all datasets that *overestimating* the mix rate causes substantial degradtion sin performance, while *underestimating* the mix rates causes less damage, and in some cases actually benefits performance. This supports our hypothesis that in lieu of access to the ground truth mix rate, it is preferable to assume a *lower-bound*, at the (potential cost) of sample efficiency. In fact, actually surprising to us that assuming a lower bound hinders performance *all all*, as we would have expect performance to be relatively constant as the fraction of disagreeing instances increased, and only degrade *relative* to methods assuming the higher ground truth mix rate (though we do see something like this pattern in Waterbirds, as the absolute performance of ACE 0.1 stays roughly constant, but its performance gap with ACE 0.5 shrinks as the mix rate increases).

Overall, our results show that ACE tends to outperform DivDis when features are not statistically independent, especially under higher target mix rates.

### 4.3 LEARNING LOW DENSITY SEPARATORS

Inspired by the original pseudo-labeling paper that found pseudo-labeling compresses class condition distributions, we were curious whether ACE induced a similar phenomenon. To test this hypothesis, we plot the first two principle components of the the final layer activations of the held-out test set before and after ACE training. Sample plots are show in Figure **??**. In general, we observe that ACE further separates class distributions that were already linearly separate, and disentangles class distributions that previously had strong overlap. This result supports the idea that ACE tries to learn low density separators, and also illustrates a benefit of predictive diversity over representational diversity (e.g. (Teney et al., 2022)) - predictive diversity can learn representations relevant to the target domain that may be otherwise absent from a generically pre-trained model.

### 4.4 GOAL GENERALIZATION IN DEEP REINFORCEMENT LEARNING

To evaluate question 6, we use ACE to learn diverse reward functions on the CoinRun deep reinforcement learning environment (Cobbe et al., 2020), with the coin randomly placed throughout the level (Langosco et al., 2023).

**CoinRun** CoinRun is a procedurally generated 2D side-scroller where the agent must avoid obstacles and cliffs and collect the gold coin at the end (far right) of the level. (Langosco et al., 2023) finds that randomly placing the coin causes a significant drop in agent performance, as the agent learns to move right rather than collect the coin. We reformulate Coinrun as a binary image classification problem, concatenating images $x$ from agent trajectories at time $t-1, t$ and labeling based on whether the agent received reward at time $t$:$h(x_{t-1}, x_t) = R(s_t)$ (see Figure **??** for an example). Agent trajectories are sampled from an agent programmed to randomly move right. The source distribution contains agent trajectories and rewards from the standard CoinRun environment with the coin placed at the end of the level. The target distribution contains trajectories on environments where the coin in randomly placed (not using reward data, in keeping with the unlabeled target distribution assumption). See Appendix E for more details on the experimental setup.

We train two heads $h_1, h_2$ with ACE on the constructed CoinRun dataset. Using the learned heads, we then train deep reinforcement learning agents with two configurations. The first, "Prudent Agent", is trained using the average of the two heads as a reward: $R(s_t, a_t) = \frac{h_1 + h_2}{2}$. For the second, "ACE Agent", we sample a target instance on which the two heads strongly disagree, and use it to select one of the classifiers as the reward function. If the ground truth reward is 1, we pick the classifier with the higher prediction as the reward function, else we pick the classifier with the lower prediction: $R(s_t, a_t) = h_{coin}$. This process enabled us to distinguish the ground truth (Coin) and spurious (Right) features using one bit of human feedback (Note this is exactly the "Disambiguate" step in (Lee et al., 2023)). On a set of 1,000 levels not seen by the ACE or agent training process with the coin randomly placed, we find the Prudent Agent significantly outperforms the standard agent, and that the ACE agent significantly outperforms the Prudent Agent. See Figure 3 for a plot of agent performance.

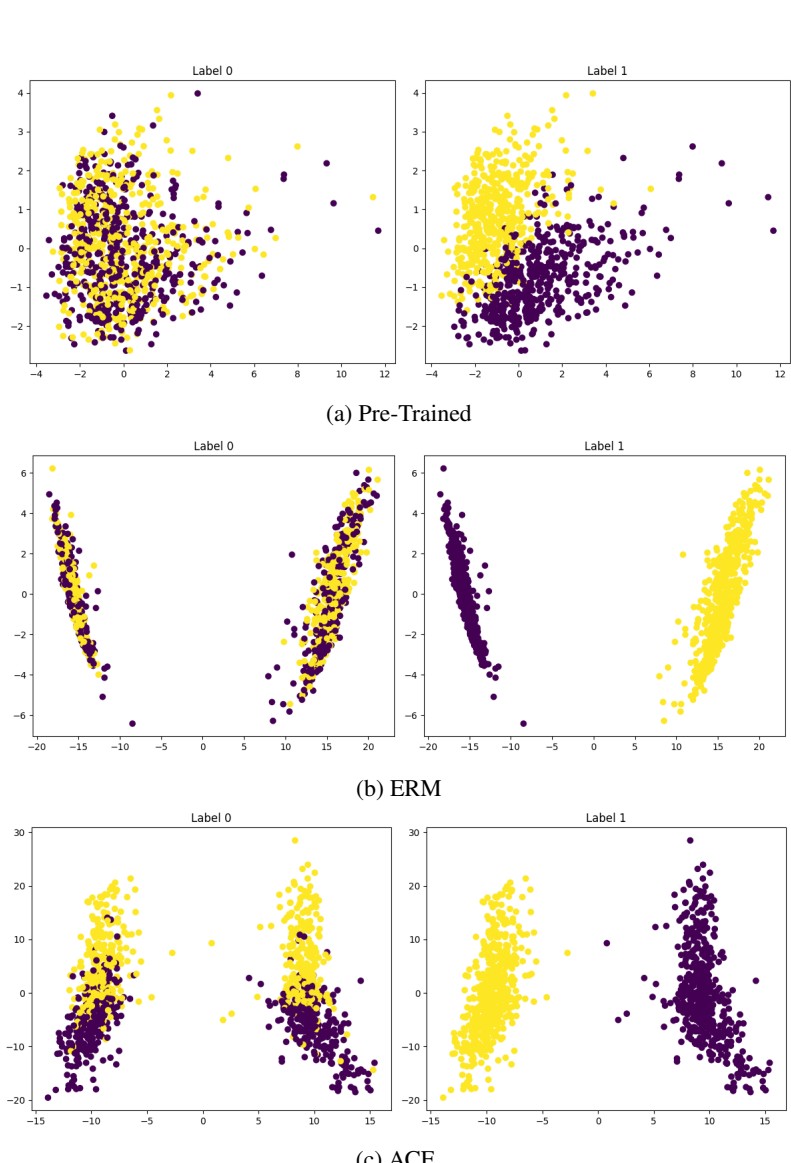

Figure 2: **ACE Learns Low Density Separators** The first two principle components of activations on the CIFAR-MNIST test set taken a) before any finetuning, b) after standard emperical risk minimization on the source distribution, c) after training with ACE. Label 0 (left) is the CIFAR label (cars vs trucks) and label 1 (right) is the MNIST label (0's and 1's). Purple corresponds to the 0 label *within each feature* (i.e. 0's and cars) and yellow corresponds to the 1 label (i.e. 1's and trucks). Both ERM and ACE separaate the MNIST classes, but only ACE separates the CIFAR classes.

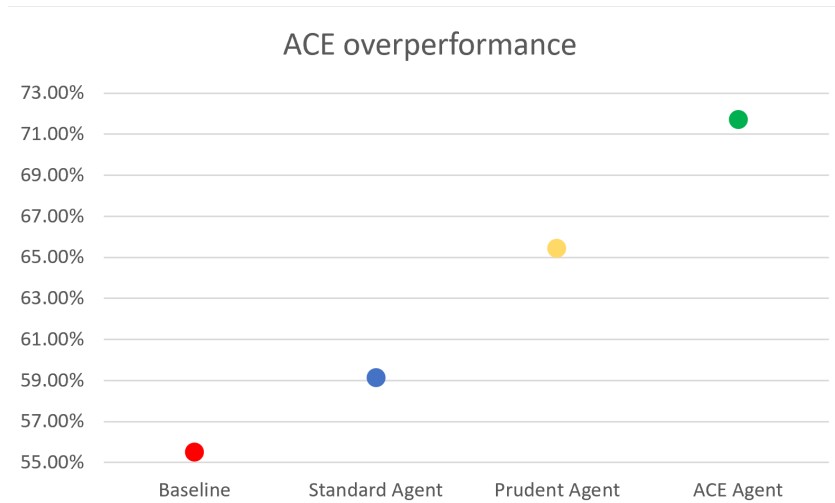

Figure 3: **Performance on CoinRun** 'Baseline' is a simple policy of 'always go right'. 'Standard Agent', trained on labeled training data, goes right while also avoiding monsters and holes, and sometimes collecting the coin on the way. Without additional reward information, 'ACE-Prudent Agent' learns to disambiguate getting the coin from going to the right and tries to achieve both goals. The 'ACE Agent' presents informative images of two possible reward functions, getting one bit of human feedback on which is correct.

## 5 CONCLUSION

While prior diverse generalization methods make particular (but implicit) assumptions about the target distribution, our algorithm ACE treats diverse generalization as a semi-supervised learning problem, taking a pseudo-labeling approach which flexibly incorporates information about the target distribution while still performing well under weak assumptions. We show that ACE competitive is more robust than DivDis to variations in the target distribution *mix rate* (the proportion of instances with disagreeing labels that "break" spurious correlations) showing particularly strong gains under mix rates higher than 0.5. We also demonstrate an initial application of ACE to alignment, using it to mitigate goal misgeneralization in deep reinforcement learning.

### 5.1 LIMITATIONS AND FUTURE WORK

In the present work, we mostly assume access to the ground truth target mix rate (except when assuming lower bounds), and always assume access to the number of spurious correlates / features. While we expect to be able to make such assumptions in many applications, future work could explore dynamically learning both. We have also seen preliminary success using ACE to detect adversarial images, and under-specification in text data, and consider both interesting future directions. Finally, we see strong analogies between ACE and recently proposed methods in weak-to-strong-generalization, and are excited about future work applying ACE to this and related problems like measurement-tampering and reward-hacking.

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

## A   RELATED WORK

**Domain Generalization** attempts to generalize from the training distribution to an unseen target distribution, typically leveraging labeled data from multiple domains that vary along a spurious attribute (Arjovsky et al., 2020; Chevalley et al., 2022; Sagawa et al., 2020; Krueger et al., 2021). Without access to unlabeled target distribution data, domain generalization methods must rely on sufficiently diversity in the training distribution. However, when sufficient diversity exists, standard expected risk minimization can outperform domain-generalization methods (Gulrajani & Lopez-Paz, 2020).

**Domain Adaptation** assumes access to unlabeled target data, but is less directly concerned with resolving spurious and more concerned with handling distributional shifts that effect all data instances such as the type of camera used to capture images (Saenko et al., 2010; Wilson & Cook, 2020). Given this focus on general domain shifts rather than spurious correlations, it makes sense that prominent domain adaptation methods such as Deep-CORL (Sun & Saenko, 2016) and MMD-VAE (Li et al., 2018) tend to perform poorly on spurious correlation benchmarks (Tiwari & Shenoy, 2023; Lynch et al., 2023).

**Diverse Ensembles** have been used in different ways to try to circumvent spurious correlations and simplicity bias. Learning with Biased Committee (Kim et al., 2023) uses an ensemble with models to weight samples for the training of a final classifier, emphasizing samples with less consensus. Each classifier in the ensemble is initialized and trained differently, and the authors assume this will be sufficient for them to be diverse, even though no explicit penalty is considered to enforce it.

(Teney et al., 2022) trains multiple models in parallel, with a diversity regularize to make them represent different functions. They penalize the alignment of gradients of the different models. This idea is similar to ACE in essence, but unable to leverage unlabeled data and relies on gradient alignment corresponding to representational diversity.

Most similar to ACE, Diversify and Disambiguate (DivDis)(Lee et al., 2023) and Diversity by Disagreement Training (D-BAT) (Pagliardini et al., 2022) train classifiers to output diverse predictions on the target data, DivDis by penalizing mutual information between the classifiers and D-BAT by maximizing disagreement. However, both methods make strong distributional assumptions about the target dataset: DivDis assumes an equal proportion of in-distribution and correlation-breaking instances, and D-BAT (implicitly) assumes all instances break spurious correlations.

**Hard Example Mining** is the technique to gradually grow, or bootstrap, the set of examples by selecting those for which the detector triggers a false alarm (Shrivastava et al., 2016). It is used in fairness applications to identify and mitigate adversarial examples (Lin et al., 2023).The focal loss (Lin et al., 2017) was designed with a similar objective of emphasizing hard examples and preventing the vast number of easy examples to dominate training. The exponentially-decaying sum of losses introduces with ACE has an analogous objective to these techniques of emphasizing the examples where the classifiers differ the most.

**Continuous Learning (CL)** is the ability of AI to continually acquire, update, accumulate and exploit knowledge (Wang et al., 2024). This knowledge comes from incrementally ingesting new data that comes from a dynamic data distribution. (Lesort, 2023) points that one of the challenges in CL is to discover causal relationships between features and labels under distribution shifts, and that for that it needs to deal with spurious correlations to learn robust representations. By solving spurious correlations, ACE can be applied to improve performance under the natural distribution shifts.

## B  IMPLEMENTATION

Below in an implementation of ACE for binary clasification and two heads:

```python
import torch

def topk_loss(logits, criterion=torch.nn.functional.binary_cross_entropy_with_logits,
              r_01=0.25, r_10=0.25):
    """
    Args:
        logits (torch.Tensor): Input logits with shape [BATCH_SIZE, HEADS].
        criterion (torch.nn.Module): Loss criterion.
        r_01 (float): Expected proportion of samples labels (0,1).
        r_10 (float): Expected proportion of samples labels (1,0).
    """
    # get batch size
    bs = logits.shape[0]
    # compute head losses
    head_0_0 = criterion(
        logits[:, 0], torch.zeros_like(logits[:, 0]), reduction='none'
    )
    head_0_1 = criterion(
        logits[:, 0], torch.ones_like(logits[:, 0]), reduction='none'
    )
    head_1_0 = criterion(
        logits[:, 1], torch.zeros_like(logits[:, 1]), reduction='none'
    )
    head_1_1 = criterion(
        logits[:, 1], torch.ones_like(logits[:, 1]), reduction='none'
    )
    # compute disagreement losses
    loss_0_1 = head_0_0 + head_1_1
    loss_1_0 = head_0_1 + head_1_0
    # sort losses in ascending order
    loss_0_1, _ = loss_0_1.sort()
    loss_1_0, _ = loss_1_0.sort()
    # compute top k losses
    k_01 = round(bs * r_01)
    k_10 = round(bs * r_10)
    loss_0_1 = loss_0_1[:k_01].mean()
    loss_1_0 = loss_1_0[:k_10].mean()
    # compute total loss
    loss = loss_0_1 + loss_1_0
    return loss
```

# C  EXPONENTIALLY-WEIGHTED TOP-K ACE

Note that even with ground-truth access to the joint class distribution, we still expect there to be noise in the ordering of the instances according to hypothesis probabilities. Thus, to attempt to avoid cascading effects where initial confidently wrong predictions throw off the learning process, we apply a heuristic exponential-decay re-weighting to the instance losses, smoothly approximating the top-k loss. Concretely, the exponentially smoothed loss is given by:

$$\mathcal{L}_e(h_1, \ldots, h_H) = \sum_g \frac{\lambda_g}{k_g} \sum_{j=1}^{N} \sum_{i=1}^{H} e^{-(i-1)} L(h_i(x_j), g_i) \tag{6}$$

We provide no formal defense of this exponential smoothing, and we would not be surprised if other more principled smoothing methods performed better [4]. In practice though, the exponential smoothing tends to out perform the default (top-k) ACE.

A significant limitation of exponential smoothing is that we cannot set the weighting based on the expected feature distribution. To overcome this limitation, we treat the exponential weighting as approximating top-k with k roughly in the range [2,8] depending on the task, and (assuming an even distribution of disagreeing instances such that e.g. $p^{(0,1)} = p^{(1,0)}$), set the target batch size based on the expected mix rate: $N = \frac{k}{r}$.

---

[4]Early experiments have explored probabilistic weighting based on the distribution of group labels in a batch, while approaches might try modeling noise the sorting of instances

## D    RELEVANCE TO AI ALIGNMENT

We believe resolving spurious correlations is underappreciated in the context of AI Existential Safety and Alignment ((Carlsmith, 2024; Critch & Krueger, 2020; Ngo et al., 2024; Bengio et al., 2024)). Many common worries in AI safety and ethics can be reformulated as under-specification problems, which we enumerate below.

**Sycophancy** Supervisor approval is a spurious correlate for supervisor approval *given the supervisor knows everything the model knows* ((Cotra, 2021; Sharma et al., 2023; Denison et al., 2024)). Caveats: Source distribution will contain mislabeled examples.

**Reward Tampering** The reward itself is a spurious correlate of the outcome or behavior being rewarded *reward tampering/wire-heading* ((Everitt & Hutter, 2016; Kumar et al., 2020; Everitt et al., 2021; Denison et al., 2024)). Caveats: Source distribution will contain mislabeled examples.

**Measurement Tampering** The measurements are spurious correlate of the ground-truth entity being measured (Roger et al., 2023). Caveats: None.

**Weak to Strong Generalization** Weak labels are a spurious correlate for ground-truth answers (Burns et al., 2023). Caveats: Source distribution will contain mislabeled examples.

**Scheming** Time-independent goals are a spurious correlate for episode-dependent goals (Carlsmith, 2023). Caveats: Questionable whether goals will be cleanly separable such that we can learn diverse generalizations.

This is not a comprehensive list, and some examples are more applicable than others. In general though, we think methods for to a) detect when a spurious correlation present on the training distribution has been broken and b) learning all plausible "resolutions" of the spurious correlation, would be very useful for alignment. We are particular excited to test ACE on measurement tampering (Roger et al., 2023), as the setting nearly identical to the complete spurious correlation setting studied here, with some a-priori knowledge of the target distribution (no instances where the measurements are off but the ground truth feature is present).

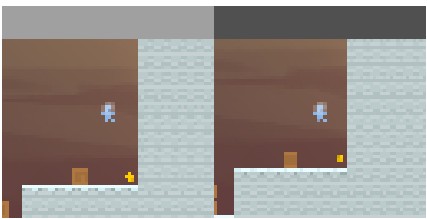

Figure 4: Example of a victory state

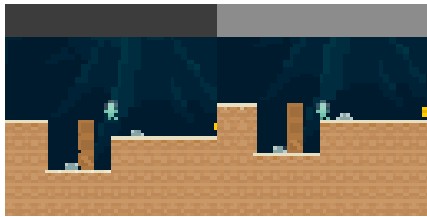

Figure 5: Example of a non-victory state

# E  COINRUN DETAILS

## E.1  MOVE-RIGHT AGENT

The Move-Right agent follows a fixed action sequence: 8,4,4,-1,7,-1, with 8 meaning 'jump right', 4 meaning 'do nothing', 7 meaning 'move right' and -1 meaning 'select a random action from 0 to 8 inclusive'.

## E.2  LABELED TRAINING DATA

That 'move-right' agent was deployed on environments where the coin was on the right of the level. When the agent reached the coin, it received a reward and the level ended. Since it received a reward, this data was 'labeled'. This was run $10,000$ times, and we selected randomly $5,000$ victory states (images of the game screen just before the agent won) and $5,000$ non-victory states (images of the game screen at any other point).

The action chosen was encoded as a grey-scale bar atop the image. Because the CoinRun game has momentum from turn to turn, we always collected an image along with the one just before it.

## E.3  UNLABELED ENVIRONMENT DATA

The 'move-right' agent was also deployed in environments where the coin was placed randomly in the level. It generate $50$ different runs of $50$ steps each (unless the agent died beforehand). If the agent ever hit the coin or the right of the level, the run would not end, and there would be no reward information shared with the agent, or anything to indicate that this state was special.

All in all, this generated a total of $2,366$ different images (in theory there should have been $49$ images per run – since each image has a state and the one before it – for a total of $50 \cdot 49 = 2,450$ images, but some of the runs were cut short when the agent was eaten by a monster). Of there, $84$ were images of a win state where the agent was getting the coin; however, this information was not made available to the training process.

## E.4  TRAINING THE AGENT

We follow the same procedure as (Langosco et al., 2023), using Proximal Policy Optimization (Schulman et al., 2017) and the same hyper-parameters.

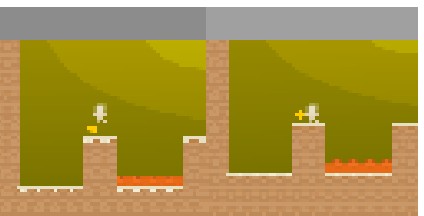

Figure 6: Example of an unlabeled victory state

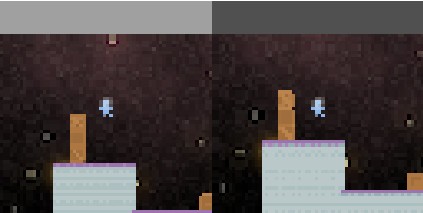

Figure 7: Example of an unlabeled non-victory state

### E.5 EVALUATING AGENT PERFORMANCE

We found standard agent performance (performance of agent trained on levels with the coin at the end on levels with the coin randomly placed) to be significantly higher than performance reported by (Langosco et al., 2023)( approximately 60% vs 25%). Given that we used the code provided by (Langosco et al., 2023) to replicate the results, we believe this to a bug in the reporting of the original results, an assessment which the corresponding author agrees is plausible.

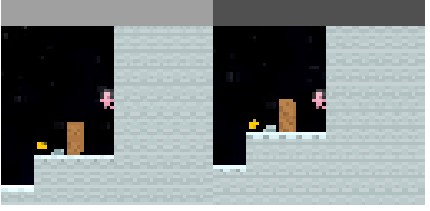
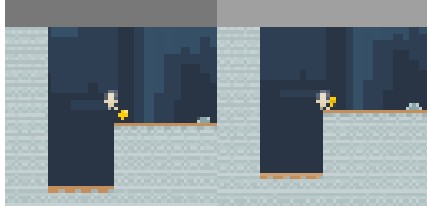

(a) High score on $R_0$.      (b) High score on $R_1$.

Figure 8: Indicative images of high scores for the two candidate functions.

