# OpenReview forum: "Algorithm for Concept Extrapolation: Diverse Generalization via Selective Disagreement"
_ICLR.cc/2025/Conference — Submitted to ICLR 2025_

### Official Review · Reviewer_i8aL · 2024-11-01

**Soundness:** 3
**Presentation:** 2
**Contribution:** 2
**Rating:** 5
**Confidence:** 3

**Summary:**

The paper presents ACE, algorithm for concept extrapolation, which uses unlabeled data from the target distribution to learn more robust models. The key idea is to learn to diverse classifiers on the source distribution which disagree on the correlation-breaking instances from the target distribution. Empricailly, ACE performs well on two datasets containing spurious correlations and also in an RL setting with goal-misgeneralization. Importantly, ACE works well even with a low amount of spurious correlation breaking instances in the target distribution, unlike prior work.

**Strengths:**

1. The proposed method is principled and the paper makes interesting arguments (although more conceptual / intuition based) on: (a) why it makes more sense to focus on predictive diversity compared to representational diversity (sec 2.2) ; (b) why focusing only on correlation-breaking instances from the target dataset makes more sense than using all unlabeled examples from the target distribution.

2. The proposed method is reasonably simple, where only an additional regularization term based on disagreement is added.

3. The paper shows that the proposed method, ACE has a particular advantage over existing methods at low target mix rate.

4. The paper also shows connections between spurious correlations and goal misgeneralization/reward hacking which is interesting, and the proposed method also shows initial promise for it (Figure 3)

**Weaknesses:**

1. The baselines in the main tables (1 and 2) are hard to interpret since they use a different backbone model than the proposed method ACE. (ResNet vs ClipVIT). Given that, it’s not clear if the improvements are just due to a better backbone model or due to the proposed method.

2. Missing Ablations — one of the main claims in the paper and the main difference from prior work (DAT-B) is to only focus on correlation breaking instances in the target dataset instead of using all unlabeled examples. It would be useful to have this ablation to empirically show this.

3. The paper only focuses on the setting of complete spurious correlations i.e. the spurious and ground true feature both always agree on the source dataset. While that is fine to use for controlled setting, the paper would be much stronger if it also showed results on a realistic setting with non-zero source rate.

    a. Realistically, the true feature should perfectly explain the source dataset (up to some noise) whereas the spurious feature should not.

    b. Additionally, the paper could be much stronger if it evaluated on more naturally occurring spurious correlations rather than artificially created as in CIFAR-MNIST or M2M (e.g. MultiNLI or CivilComments-WILDS as used in https://arxiv.org/pdf/2107.09044)

4. (minor) Missing baselines — given that DivDis is a very similar method and already used in Table 1, it’s unclear why that baseline is not reported for other results such as in Table 2.

**Questions:**

1. Line 104 — does this mean the no. of classifiers required is exponential?

2. Line 100 – missing f?

3. Line 106 – missing f in f*?

4. Line 190: the the learned classifiers → the learned classifiers

5. Line 229 – missing “)”

6. Line 288 – mentions ‘four key claims’ but there seem to be only three.

7. Line 429 — ‘where the coin in placed’ → ‘where the coin is placed’

8. \citet{} vs \citep{} — in a lot of places the citation should not be in text i.e. in bracket using \citep{}.

9. Overall the paper could benefit from proofreading once more!

---

> ### Author Response · Authors · 2024-11-28
> **Common Backbone, Concerns About Complete Correlations**
>
> Thanks for the thorough feedback! Going through your concerns in order:
>
> 1. I agree having different backbones was a fundamental methodological flaw - we now use the same backbone (Resnet50) across all the image experiments
> 2. I agree we should include D-BAT in our experiments (we are planning on do shortly)
> 3. While I agree the paper would be stronger if we investigated non-zero source mix rates, the zero mix rate regime still feels pretty central, and will show up if
>
>    a) spurious correlations are sufficiently high and data collection is sufficiently constrainted
>
>    b) there are systematic factors causing spurious correlations in the labeled distribution (e.g. the provision of the reward and the behavior being rewarded in an RL context)
>
> so while I think its reasonable to want non-zero mix rate experiments, we think the paper makes a significant holist contribution without them. That said, we agree lack of real world datasets is still a significant limitation.

---

> > ### Comment · Reviewer_i8aL · 2024-12-02
> >
> > Thanks for your response, and the new experiments with the same backbone across all methods!
> >
> > Considering two reasons: (1) D-BAT is very similar to the proposed approach but is currently missing from the experiments ; (2) not much experiments with realistic spurious correlation --- either something in WILDS, or something with non-zero source mix rates which in my opinion, is more practical; I would be inclined to keep the score the same for now.
> >
> > Additionally, I do think substantial changes were made to the initial draft including complete re-working of the formulation --- considering this, I am decreasing my confidence score since I have not read the updated draft as carefully as the original draft (also would be useful to highlight what were the changes made e.g. by highlighting).

---

> > > ### Author Response · Authors · 2024-12-02
> > >
> > > makes sense, hoping to include those experiments soon. Also fyi the revised formulation is mostly in 2.1 and 3 (see 2.2 for differences with D-BAT)

---

### Official Review · Reviewer_r7vF · 2024-11-03

**Soundness:** 3
**Presentation:** 3
**Contribution:** 3
**Rating:** 6
**Confidence:** 3

**Summary:**

This paper introduces a new method for handling out-of-distribution generalization in deep learning, particularly when spurious correlations in training data are not present in a new evaluation domain. The core idea is that disagreement on testing data between models that solve the training data might be a good proxy for when an example is ``hard’’ because the correlations in training break down. They leverage this signal to train a classifier on a loss that encourages disagreement on examples where correlations in training break down, but agreement on other test examples. This method achieves better performance on out of domain generalization evaluations, including a goal misgeneralization benchmark task.

My understanding is that maximizing disagreement on test examples is not new and that is the core idea of the baseline disdiv. The new idea is that disagreement should only be maximized on examples that are correlation breaking, and the disagreement of many trained classifiers could be a signal for when correlation breaking occurs.

This is not my domain of expertise, so I am not confident about whether they evaluated all relevant baselines, chose good benchmarks to evaluate on, or made good assumptions.

Use /citep and not /cite when, all of these citations are formatted incorrectly.

I think there are some typos in the math notation and I think another pass or two is necessary to get this up to make sure the rigor is uniform. In particular, try to be clear what the difference between a feature and a classifier is in the set up-- my understanding is that features and classifiers have the same type signature, i.e., something that maps from X to Y. Also consider trying to factor out a layer of superscript/subscript in the notation, I know that’s hard, but some of the terms are rough. That being said, the core ideas are communicated and I can see that the set up and method sections aim for a high level of rigor, which I appreciate.





Typo in formulation, the definition of similarity between f and g is missing an f line 100

Like 107, a lone * should by f*

The definition of the ground truth feature and spurious features suddenly becomes much less rigorous and technical. I think you should maintain the level or rigor here. What is a feature, what is correlated to what?

Line 288 four key claims, only three claims stated though

**Strengths:**

See review

**Weaknesses:**

See review

**Questions:**

see review

---

> ### Author Response · Authors · 2024-11-28
> **Simplified Notation and Formulation**
>
> Thanks for the feedback, I agree the original formulation/notation was pretty janky. We've totally reworked it, now formulating diverse generalization as a semi-supervised learning problem, and hopefully things feel a lot cleaner and more straight forward.
>
> Also thanks for the \citep catch/tip

---

### Official Review · Reviewer_NFkq · 2024-11-03

**Soundness:** 2
**Presentation:** 3
**Contribution:** 3
**Rating:** 6
**Confidence:** 4

**Summary:**

The authors introduce the Algorithm for Concept Extrapolation (ACE), which aims to learn multiple classifiers that maintain high accuracy on source distribution data while strategically disagreeing on target distribution instances that violate spurious correlations. The key idea is an exponentially-weighted disagreement loss that encourages classifier disagreement primarily on instances likely to break correlations, combined with an approach to batch size selection based on expected "mix rates" (the frequency of correlation-breaking instances). The authors provide theoretical motivation through a feature interpolation model and argue that access to unlabeled target data makes the problem more tractable than trying to learn all possible generalizations from source data alone.

The paper validates ACE through a battery of experiments showing superior performance to prior methods across multiple benchmarks. On standard spurious correlation tasks (WaterbirdsCC, CIFAR-MNIST), ACE achieves state-of-the-art results. More impressively, on the challenging Spawrious M2M-Hard dataset involving multiple interacting spurious correlations, ACE achieves 92.47% accuracy compared to the previous best of 68.93%. The authors also demonstrate ACE's robustness to varying mix rates through experiments on a custom HappyFaces dataset, showing it outperforms DivDis across all mix rates tested.

**Strengths:**

- The paper offers a clear motivation for why access to unlabeled target data makes the problem more tractable than trying to learn all possible generalizations. It then goes on to clearly formulate how spurious correlations manifest in target distributions through "mix rates". The authors also offer a thoughtful analysis of why previous approaches (like DivDis) might fail when their distributional assumptions are violated, situating the paper well in literature.
- The exponentially-weighted disagreement loss is a clever way to focus on likely correlation-breaking instances. The batch size analysis further provides practical guidance for implementation.
- The method is evaluated across a variety of tasks and compared with multiple baselines. Strong experimental results bolster the validitiy of the method on both classification and RL tasks. Results suggest clear improvement over baselines across different mix rates. The authors also offer careful ablation of mix rate effects with different batch sizes.

**Weaknesses:**

- The strong experimental results come with important caveats: the method requires target validation data for hyperparameter tuning, uses different backbone networks than baselines, and requires careful batch size selection based on expected mix rates that may be difficult to estimate in practice. The authors also claim broader relevance to AI alignment problems like reward tampering and sycophancy, though these applications remain largely theoretical.
- There are some methodological concerns as well. For instance, there is no statistical significance testing or error bars reported. There is a heavy reliance on artificially constructed datasets (HappyFaces) rather than real-world data and it appears to me that HappyFaces dataset was created using facial expression detection that likely has its own biases.
- The paper could benefit from more extensive ablation studies isolating impact of different components.

**Questions:**

See Weaknesses.

---

> ### Author Response · Authors · 2024-11-28
> **Standard Hyperparameters, Principled Model Selection, Error Bars and More Mix Rate Experiments**
>
> Thanks for the feedback! Going through your concerns in order:
>
> 1. We reworked the core image experiments to all use Resnet50 with standard (non-tuned) hyperparameters (lr=1e-4, weight_decay=1e-4), and recomputed baselines (DivDis) using our setup (in the case of CIFAR-MNIST finding performance much stronger than previousy reported). We also now use validation loss as a model selection criteria for both methods, which does not make use of target labels.
>
> 2. I think are claims to broader relevance to alignment are sufficiently caveated and are not central to the main text - they are core to the original motivation for our work, but not core to the arguments of the paper.
>
> 3. We have added error bars for all results and completely removed HappyFaces, replacing it with comparable mix rate experiments across 4 widely used datasets.
>
> 4. The most significant piece we could ablate was probably the exponential reweighting, and we actually ended up removed it entirely from the main text of the paper.

---

> > ### Comment · Reviewer_NFkq · 2024-11-28
> > **Thanks for the response and updates**
> >
> > I appreciate the updates to the paper. Is there a third dataset/task that can replace Happyface? I'd be happy with a vision or a language dataset that you can also evaluate on. Perhaps a language dataset, such as the sentiment data here: https://github.com/acmi-lab/counterfactually-augmented-data? You'll find that in the sentiment analysis dataset, most reviews for romantic movies are positive and for horror are negative. That could be a good dataset.

---

> > > ### Author Response · Authors · 2024-12-02
> > >
> > > We're currently integrating Multi MLI and CelebA (both used in prior work on spurious correlations), but the counterfactually augmented data also seems good!

---

### Official Review · Reviewer_nQKb · 2024-11-05

**Soundness:** 2
**Presentation:** 3
**Contribution:** 2
**Rating:** 5
**Confidence:** 4

**Summary:**

This study proposes a novel method, termed ACE, to address the issue of pseudo-correlation under distribution shifts, which is a challenging problem in the field of machine learning.

------

Thank the authors for their response and the additional experiments! I still believe that the paper should be further enhanced regarding clarity, especially in terms of the presentation of the algorithm's practicality and complexity. Additionally, the paper could benefit from more comparative baselines.

**Strengths:**

1. The performance improvements of the ACE algorithm compared to existing techniques are demonstrated across multiple benchmark tests.
2. The paper highlights the robustness of the ACE algorithm to variations in target distribution mixture rates, which is an important characteristic.

**Weaknesses:**

1.	Based solely on the descriptions provided in the paper, there remains some confusion regarding the application of this method. The paper should include an algorithm or pseudocode to clarify how to utilize this algorithm to address practical problems.
2.	The lack of experiments conducted on datasets other than images, especially text classification datasets, imposes certain limitations on the applicability of this method. In fact, there are many datasets in the text domain that exhibit spurious correlations, such as the CivilComments dataset. Furthermore, even within image datasets, additional datasets, such as CelebA and IN9, should be explored.
3.	The paper does not discuss the computational complexity of this algorithm, which is an important consideration for its scalability in practical applications.
4.	Although the experimental results demonstrate effectiveness, the paper does not provide sufficient theoretical support to explain why the ACE algorithm is effective, particularly regarding the choice of the exponential weighting scheme.
5.	The paper lacks a detailed introduction to the comparative baselines. To my knowledge, many methods included in the comparison, such as CORAL, IRM, ERM, and GroupDRO, were developed prior to 2020. Additionally, more advanced methods, such as LISA (2022) and Fish (2022), should also be included in the comparisons.

**Questions:**

See weaknesses.

---

> ### Author Response · Authors · 2024-11-28
> **Revamped Formulation, Python Implementation**
>
> Thanks for the constructive feedback! Going through the weakness you highlighted in order:
>
> 1. We agree the original presentation was somewhat unclear. We have significantly reworked the formulation and method section to hopefully make things more clear, as have included python implementation of the top-k (ACE) loss
>
> 2. The lack of text datasets is certainty a shortcoming which we plan to address in the future. For now, we have added more extensive experimentation with Waterbirds
>
> 3. The algorithm scales log linearly in the batch size which we expect to represent a small fraction of the total cost of computing a forward and backward pass. Perhaps more worrisome is the exponential scaling in the number of heads, but in practice we expect this number to be fairly small (only 2 in our experiments), and importantly,  does not involve e.g. computing exponentially many forward passes
>
> 4. I agree the original formulation was not sufficiently motivated - we have significantly refined our formulation and theoretical justification, interpreting diverse generalization as a semi-supervised learning problem and ACE as a pseudo-labeling approach (essentially "importing" the standard theoretical justifications for pseudo-labeling approaches). See the revised section 2 and 3.

---

### Author Response · Authors · 2024-12-02
**Reformulation, Error Bars, More Datasets**

Thanks to everyone for their helpful feedback.

First, **due to a latex error, a substantial portion of the revised methods section was excluded**, see here: https://anonymous.4open.science/api/repo/diverse-gen-37E2/file/ACE_paper.pdf?v=7c6f92f0 for a revised version


We have addressed reviewer specific criticisms in comments for each review. Here we provide an overview of the changes and experiments we plan to add in the near future.

# Revised Formulation
We revised our problem formulation and method section to treat diverse generalization as a semi-supervised learning problem with multiple feature labels, and ACE as a pseudo-labeling approach which constructs pseudo-labels for each feature in proportion to the expected disagreement rate (and optionally the expected class distribution). These modifications both simplify the presentation of ACE and provide more rich, theoretically grounded motivations. In particular, see sections 2.1 and 3.


# More Extensive Mix Rate Experiments with Error Bars
To remove our dependence on the Happyface dataset and provide generally more robust comprehensive results, we ran ACE and DivDis on 4 benchmark datasets used in prior work across different mix rates, and generated results over three random seeds to produce error bars

# Next Steps
There are three motifications we are in the process of adding:
1. Adding CelebA and MultiNLI to our set of datasets
2. Adding experiments with D-BAT
3. Adding experiments with incomplete spurious correlations on the source distribution

While we believe these experiments would improve the paper,  the number of datasets tested is in line with prior work (i.e. DivDis, D-BAT). The paper is also centrally focused on complete spurious correlation, and while further experiments with incomplete spurious correlations would certainly improve the results, we believe the paper is still a strong novel contribution as is

---

### Meta-Review · Area_Chair_jDpA · 2024-12-19

**Metareview:**

This paper introduces a new ensembling approach to handling distribution shift by training classifiers to disagree when new data violates spurious correlations.

Pros:
Intuitive and interesting method, appears to beat all baselines. Several datatsets are tested.

Cons:
Both the improvements in presentation and the additional experiments in more realistic environments have been promised by the authors, but the results are not yet available and the presentation is still somewhat unclear. More recent baselines were requested by nQKb but the results are not yet available for that  either.

**Additional Comments On Reviewer Discussion:**

Reviewer nQKb maintains that the paper is unclear and has some presentation issues, and needs more recent baselines. The authors refer to improvements that they promise to make for future versions, but the current state maybe is not yet ready. The authors made a number of improvements such as incorporating error bars, but other promised improvements do not have results available yet.

---

### Decision · Program_Chairs · 2025-01-22

Reject